# A comprehensive meta-analysis of transcriptome data to identify signature genes associated with pancreatic ductal adenocarcinoma

Shirin Omidvar Kordshouli[1], Ahmad Tahmasebi[1], Ali Moghadam[1], Amin Ramezani[2,3]*, Ali Niazi[1]*

1 Institute of Biotechnology, Shiraz University, Shiraz, Iran, 2 Department of Medical Biotechnology, School of Advanced Medical Sciences and Technologies, Shiraz University of Medical Sciences, Shiraz, Iran, 3 Shiraz Institute for Cancer Research, School of Medicine, Shiraz University of Medical Sciences, Shiraz, Iran

* niazi@shirazu.ac.ir (AN); aramezani@sums.ac.ir (AR)

## Abstract

### Purpose

Pancreatic ductal adenocarcinoma (PDAC) has a five-year survival rate of less than 5%. Absence of symptoms at primary tumor stages, as well as high aggressiveness of the tumor can lead to high mortality in cancer patients. Most patients are recognized at the advanced or metastatic stage without surgical symptom, because of the lack of reliable early diagnostic biomarkers. The objective of this work was to identify potential cancer biomarkers by integrating transcriptome data.

### Methods

Several transcriptomic datasets comprising of 11 microarrays were retrieved from the GEO database. After pre-processing, a meta-analysis was applied to identify differentially expressed genes (DEGs) between tumor and nontumor samples for datasets. Next, co-expression analysis, functional enrichment and survival analyses were used to determine the functional properties of DEGs and identify potential prognostic biomarkers. In addition, some regulatory factors involved in PDAC including transcription factors (TFs), protein kinases (PKs), and miRNAs were identified.

### Results

After applying meta-analysis, 1074 DEGs including 539 down- and 535 up-regulated genes were identified. Pathway enrichment analyzes using Gene Ontology (GO) and the Kyoto Encyclopedia of Genes and Genomes (KEGG) revealed that DEGs were significantly enriched in the HIF-1 signaling pathway and focal adhesion. The results also showed that some of the DEGs were assigned to TFs that belonged to 23 conserved families. Sixty-four PKs were identified among the DEGs that showed the CAMK family was the most abundant

**Funding:** The authors received no specific funding for this work.

**Competing interests:** The authors have declared that no competing interests exist.

group. Moreover, investigation of corresponding upstream regions of DEGs identified 11 conserved sequence motifs. Furthermore, weighted gene co-expression network analysis (WGCNA) identified 8 modules, more of them were significantly enriched in Ras signaling, p53 signaling, MAPK signaling pathways. In addition, several hubs in modules were identified, including *EMP1*, *EVL*, *ELP5*, *DEF8*, *MTERF4*, *GLUP1*, *CAPN1*, *IGF1R*, *HSD17B14*, *TOM1L2* and *RAB11FIP3*. According to survival analysis, it was identified that the expression levels of two genes, *EMP1* and *RAB11FIP3* are related to prognosis.

## Conclusion

We identified several genes critical for PDAC based on meta-analysis and system biology approach. These genes may serve as potential targets for the treatment and prognosis of PDAC.

## Introduction

Pancreatic cancer (PC) is one of the most lethal types of cancer, with exocrine cells accounting for approximately 95 percent of cases. This type of PC is commonly known as PDAC [1], and is one of the most common malignant tumors of the gastrointestinal tract, as well as the seventh leading cause of cancer death worldwide [2], with a five-year survival rate of less than 5%. According to projections, PDAC will overtake breast and colorectal cancer as the second leading cause of cancer death by 2030. It has been observed that there is high mortality and very poor prognosis of PDAC as a result of unclear early symptoms and lack of specific molecular markers for early diagnosis. Most patients with advanced cancer are diagnosed with local invasion or distant metastasis [3]. Therefore, identification of the key genes and pathways is necessary to deepen our understanding of the molecular mechanisms of PDAC that can provide reliable biological markers and treatment targets [4].

Gene expression has also become a powerful tool in recent years for predicting the role and activity of genes. Advances in high-throughput measurement technologies and a large amount of gene expression data in public databases provide an opportunity to obtain more reliable and transparent results. Besides, using meta-analysis techniques has been increasingly employed to integrate data from different resources and is especially useful for combining several datasets related to the same disease when they are limited in size [4] for increasing statistical power [5]. In addition, exploration of interactions among genes can help to better explain the complex mechanisms of biological processes. Expression analysis identifies those genes that have similar expression patterns. Genes that express a high degree of expression are likely to be involved in a common biological process or metabolic pathway [6]. WGCNA identifies correlation patterns among the genes, detecting highly correlated gene modules and summarizing many of the hub genes and biomarkers [7]. Currently, WGCNA has been used for several types of cancer that have been associated with promising results [8].

In this study, we applied large-scale microarray data for meta-analysis to find DEGs associated with PDAC. Following that, WGCNA was used to identify the co-expression of genes. Various bioinformatic methods were also applied to help in the identification of the most important candidate genes that can be considered as potential biomarkers and therapeutic targets in PDAC.

## Methods

### Data collection

Microarray-based expression datasets were retrieved from the Gene Expression Omnibus (GEO) database (https://www.ncbi.nlm.nih.gov/) (Fig 1). To investigate transcriptome responses in pancreatic cancer, we used 11 datasets. We selected only datasets with both tumor and normal samples, which included 202 samples of normal tissue and 307 samples of tumor tissue (S1 Table).

### Datasets processing and meta-analysis

The preprocessing steps were performed on each platform independently. Affymetrix datasets were preprocessed and normalized with the Robust Multi-Array Average (RMA) approach [9] using Expression Console (Affymetrix, Santa Clara, CA, USA). The expression values of Agilent microarray data were normalized using the Loess algorithm. The probe set IDs were mapped to gene symbols according to the probe annotation files. Next, expression values of the same gene symbols were collapsed based on the mean value of each gene in each database. In addition, genes with low expression levels and low variation in expression values were removed. After processing, to identify DEGs, meta-analysis was conducted using the Rank

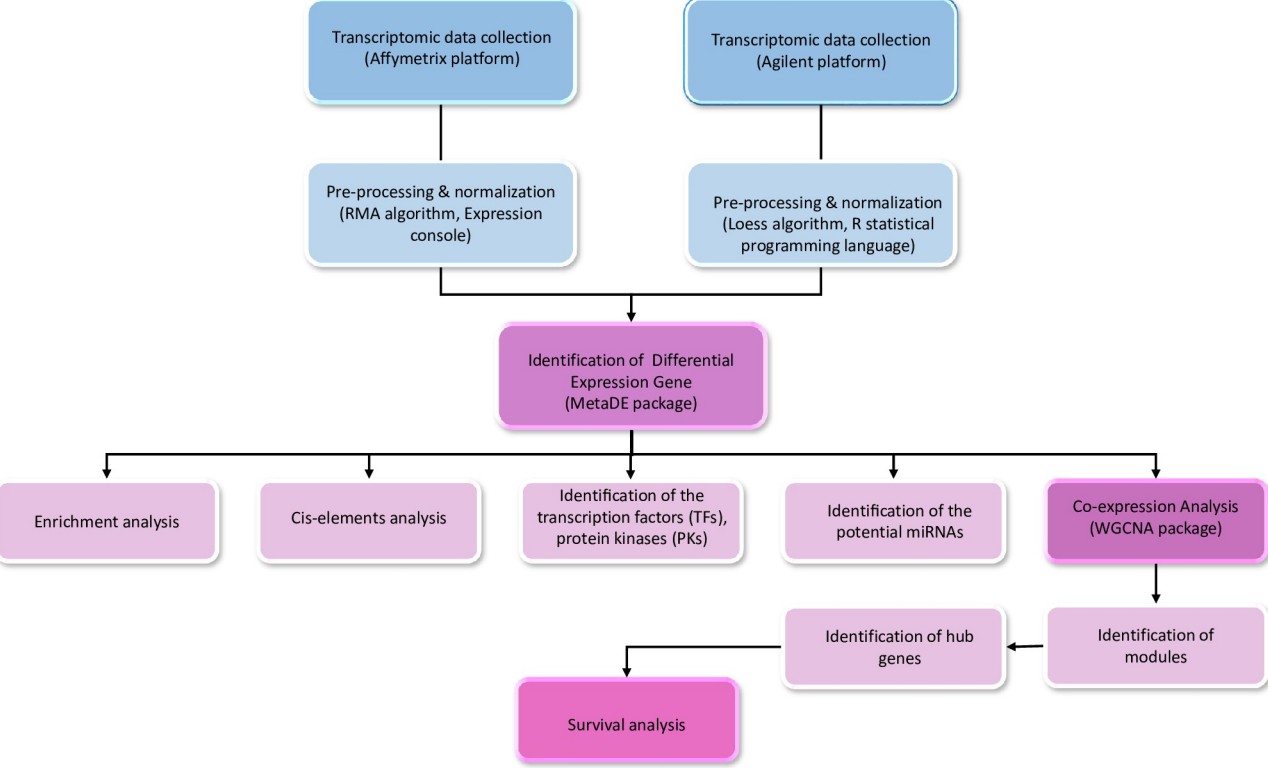

**Fig 1. Workflow including meta-analysis and bioinformatics pipeline.** Gene expression datasets of PDAC and pancreatic cancer were obtained from the GEO. The datasets were normalized and processed to identify differentially expressed genes (DEGs) between normal and tumor tissues. The significantly enriched pathways and Gene Ontology were identified through enrichment analyses. Conserved motifs and consensus cis-regulatory elements (CREs) of DEGs were detected. The WGCNA was used to cluster genes with the highest connection and identification of co-expression modules.

Prod method in the MetaDE R package [10]. Genes with a False Discovery Rate (FDR) of less than 0.05 were considered significant DEGs.

### Gene enrichment analysis

GO enrichment and KEGG pathway analyses were performed for DEGs using the g:Profiler database (https://biit.cs.ut.ee/gprofiler/gost) with an adjusted *P-value* significance level of ≤ 0.05.

### Protein-protein interaction network analysis

To investigate the interactions among the DEGs, a protein-protein interaction (PPI) network was constructed by the STRING database (http://string-db.org) with a minimum required interaction score > 0.4. The PPI network was visualized using Cytoscape (version 3.7.1) software.

### Identification of transcription factors (TFs), protein kinases (PKs) and miRNAs

Human Transcription Factors database (http://humantfs.ccbr.utoronto.ca/allTFs.php) was used to identify TFs. The identification of PKs is considered a key step towards tumor progression [11]. The GSEA database (https://www.gsea-msigdb.org) was employed to detect potential PKs among DEGs. In addition, identification of potential miRNAs that might be related to the DEGs was carried out using the miRTarBase database (http://mirtarbase.cuhk.edu.cn/php/search.php#advanced).

### *Cis*-elements analysis

CREs are one of the most important factors in regulating gene expression in various tissues and diseases. The 1500 bp upstream flanking regions of DEGs were extracted from Ensemble (https://www.ensembl.org/index.html). The MEME online tool (http://meme-suite.org/tools/meme) was used to discover conserved motifs [12]. The Tomtom tool (http://meme-suite.org/tools/tomtom) was used to define known CREs based on the motif database of JASPAR CORE 2018 [13]. The GoMo tool (http://meme-suite.org/tools/gomo) was also applied to identify possible roles and GO terms for the motifs [14].

### Weighted gene co-expression network analysis

To build the DEGs co-expression network and to identify highly correlated genes, we used the WGCNA R package [15]. Firstly, the matrix of the normalized expression values of DEGs was used to calculate Pearson's correlation coefficient between gene pairs. Next, this similarity matrix was transformed into a topological overlap measure (TOM). A hierarchical clustering tree was constructed and modules were detected with the cutreeDynamic function by cut-off a minimum module size of 30 genes. Subsequently, the network was visualized by using cytoscape software and hub genes were identified by using the cytoHubbo plug-in [16]. In addition, GO and KEGG pathway analyses of modules were conducted with DAVID under a significance threshold of FDR < 0.05 (https://david.ncifcrf.gov/).

### Survival analysis for identifying biomarker genes

To explore the potential prognostic value of hub genes, we used the Gene Expression Profiling Interactive Analysis (GEPIA) database (http://gepia.cancer-pku.cn/) for PDAC to perform an overall survival analysis using Mantel-Cox tests. Log-rank tests were used to determine

statistical significance and Log-rank P < .05 was considered significant. Furthermore, the hazard ratio (HR) was calculated based on the Cox Proportional-Hazards Model.

## Results

### Identification of DEGs

The R package MetaDE was utilized to identify DEGs. DEGs were filtered by the criterion of FDR < 0.05. Consequently, a total of 1074 DEGs, including 535 up- and 539 down-regulated genes, were identified (S2 Table). GO analysis of the DEGs revealed significant enrichment in the GO terms for 109 Biological Processes (BPs), 38 Cellular Components (CCs), 22 Molecular Functions (MFs), and also 2 KEGG pathways (adjusted *P-value* < 0.01 as cut-offs) (S3 Table). BPs include regulation of catalytic activity, metabolism of proteins, cellular response to organic substances, regulation of developmental processes, regulation of cell migration, anatomical structure morphogenesis, regulation of molecular functions, regulation of cellular component movements, and regulation of cell motility. The MFs of DEGs are mainly concentrated on protein binding, enzyme regulator activity, catalytic activity, acting on a protein, protein kinase activity, and protein serine/threonine kinase activity. The top and significant enriched KEGG pathways were HIF-1 signaling pathway and focal adhesion (Table 1).

### Identification of the TFs, PKs, and miRNAs

During tumorigenesis, some transcription factors cause overexpression or suppression of target genes, as well as changes in the biology of cancer cells. As a result, targeting of transcription factors is a possible strategy for cancer therapy. Among DEGs, 152 TFs belonged to 23 conserved families were identified, whereas 2 families were the largest groups. One is unknown and contains 73 genes and the other is *C2H2 ZF* with 29 genes (Fig 2). Sixty-four protein kinase gene were identified and classified into 12 families. The *CAMK* was the largest group among these families (Table 2 and S4 Table). A total of 39 and 25 PKs were up- and down-regulated, respectively.

We investigated potential miRNAs that may be related to DEGs using the mirtarbase database. A total of 901 miRNAs, which belonged to 195 families, were found. Among the detected miRNAs, the hsa-miR-200 family comprised the highest frequency with 59 members (Table 3 and S5 Table).

### *Cis*-elements analysis

Conserved motifs and consensus CREs were detected by analyzing the 1500 bp upstream flanking regions. Eleven significant motifs were detected by the MEME database (Table 4) and further motif enrichment was performed by using the GOMO tool (S6 Table). Gene ontology indicated that these motifs participated in sensory perception of smell, DNA damage checkpoint, regulation of organ growth, translational elongation, and RNA processing. These motifs were also involved in molecular functions such as olfactory receptor activity, structural constituent of ribosome, transcription factor activity, histone binding, proton-transporting ATPase activity, and rotational mechanism (Table 4).

**Table 1. KEGG pathways that are associated with differentially expressed genes (DEGs).**

| Pathway | Gene count | Adjusted *P-value* |
|---|---|---|
| HIF-1 signaling pathway | 22 | 0.0002 |
| Focal adhesion | 30 | 0.0015 |

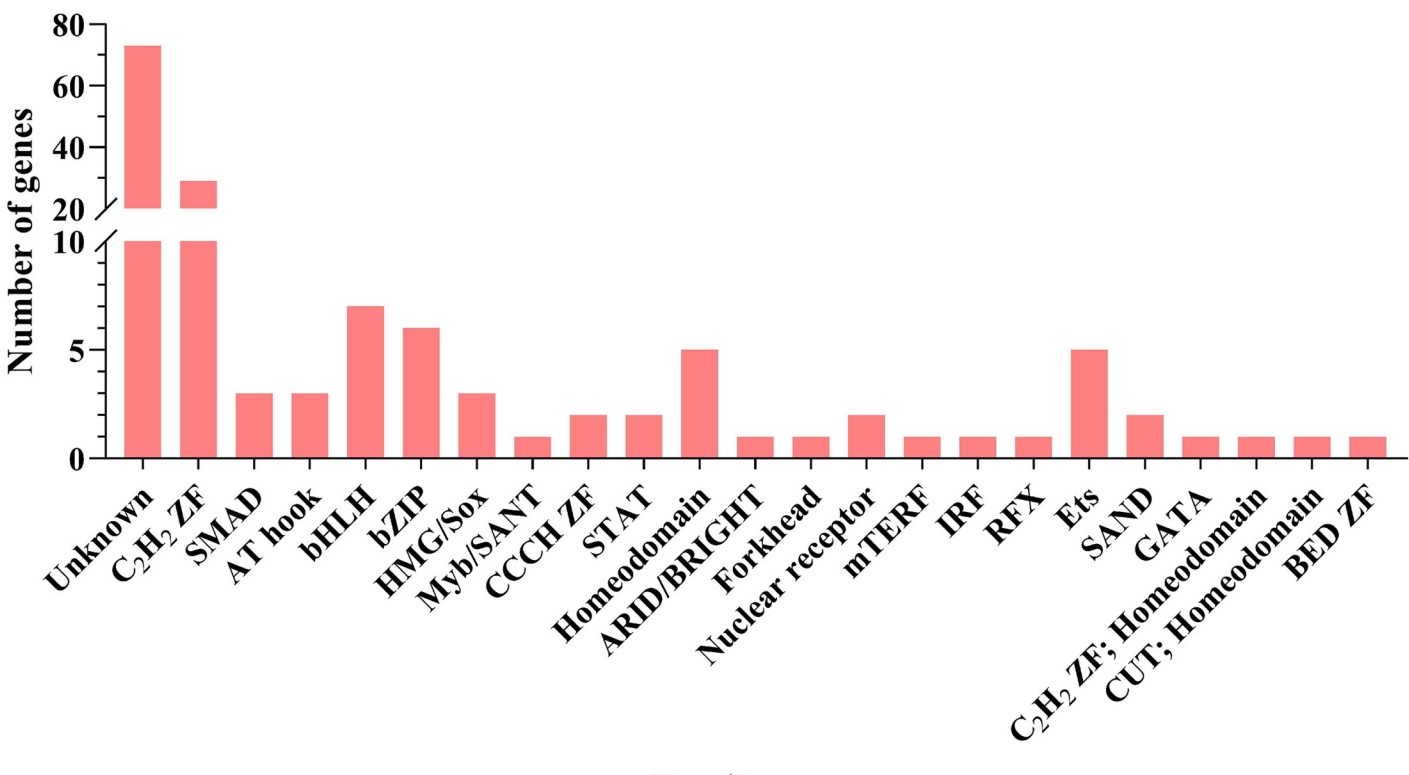

**Fig 2. Number of differentially expressed genes (DEGs) in each TF family.**

**Table 2. List of protein kinase families.**

| Family | Number of genes | Genes |
|---|---|---|
| Tyr | 7 | *ABL2, EGFR, ERBB3, FGFR1, IGF1R, INSR, DDR2,* |
| TKL | 3 | *BMPR2, TGFBR1, RIPK4* |
| CAMK | 12 | *CAMK2G, DAPK3, MKNK2, MYLK, STK11, TRIO, MKNK1, CASK, STK17B, SIK2, PRKD2, CAMK1D* |
| STE | 11 | *MAP3K8, PAK1, PAK3, STK3, STK4, STK10, MAP3K12, MAP3K13, MAP4K4, TNIK, STK26* |
| CMGC | 8 | *CDK1, CDK6, EFEMP1, MAPK6, MAPK9, HIPK2, NLK, CLK4* |
| AGC | 4 | *PRKCE, CDC42BPA, CIT, MAST4* |
| CK1 | 1 | *CSNK1G3* |
| Atypical: PI3/PI4-kinase family | 2 | *ATR, PIK3CA* |
| Atypical: PDK/BCKDK protein kinase family | 2 | *PDK1, PDK4* |
| Atypical: Alpha-type protein kinase family | 1 | *ALPK1* |
| Atypical: RIO-type Ser/Thr protein kinase family | 1 | *RIOK3* |
| Other | 12 | *PLK3, ERN1, EIF2AK2, RNASEL, PPM1D, TGFBR3, HTATIP2, EIF2AK3, ERN2, MMD, WNK1, PEAK1* |

**Table 3. Twenty-five top mirRNAs that target differentially expressed genes (DEGs).**

| miRNA Family | Count of Genes |
|---|---|
| hsa-miR-200 | 59 |
| hsa-let-7 | 36 |
| hsa-miR-29 | 30 |
| hsa-miR-34 | 26 |
| hsa-miR-125 | 24 |
| hsa-miR-133, hsa-miR-7 | 18 |
| hsa-miR-26, hsa-miR-16, hsa-miR-146, hsa-miR-27 | 17 |
| hsa-miR-155 | 15 |
| hsa-miR-124, hsa-miR-181 | 14 |
| hsa-miR-145, hsa-miR-15, hsa-miR-223 | 13 |
| hsa-miR-203, hsa-miR-30 | 12 |
| hsa-miR-21 | 11 |
| hsa-miR-107, hsa-miR-137, hsa-miR-199 | 10 |
| hsa-miR-375, hsa-miR-23 | 9 |

## WGCNA and identification of modules

We used the WGCNA approach to examine gene co-expression patterns in pancreatic cancer mRNA expression profiles. Initially, a similarity matrix was calculated based on Pearson correlation between each DEG pair, which was converted to a proximity matrix using a power function (β). Then, the topological overlap matrix (TOM) was calculated for hierarchical clustering analysis. Finally, a dynamic tree cutting algorithm was implemented to identify gene expression modules. The parameters used in this study were β power of 12 and a minimum module size of 30. Eventually, the DEGs were based on the dynamic tree cutting algorithm grouped into eight modules, which were labelled by different colors (turquoise, brown, blue, black, yellow, red, green, and pink). The modules ranged in size from 80 (pink module) to 256 (turquoise module) genes (Fig 3 and S7 and S8 Tables).

To understand the biological functions associated with modules, the enrichment analysis for the BP category was conducted by using DAVID (FDR< 0.05). The results revealed that modules were more involved in the Ras signaling pathway, p53 signaling pathway, MAPK signaling pathway, protein processing in endoplasmic reticulum, proteoglycans in cancer, focal adhesion, Rap1 signaling pathway, PI3K-Akt signaling pathway, HIF-1 signaling pathway, FoxO signaling pathway, ErbB signaling pathway, and insulin signaling pathway (Fig 4 and S9 Table). Among the modules, the highest number of TFs belonged to the turquoise module, with 22 TFs, which can indicate the regulatory role of this module (Fig 5). Moreover, a total of 11 PKs were identified in the turquoise module (S10 Table).

## Hub genes analysis in modules

To identify genes with central roles in the network, Cytoscape's CytoHubbo plugin was used to select genes with high connectivity within each module, known as hub genes. In the CytoHubbo plugin, the top 10 nodes calculated by the maximal clique centrality (MCC) algorithm were shown as hub genes in each module. Finally, from eight modules, 80 hub genes were selected (S11 Table). The hub genes were found to be significantly enriched in 12 GO terms and 4 pathways (S12 Table). These hub genes were enriched in negative regulation of adiponectin secretion, and cell division. The turquoise module yielded the highest number of hub

**Table 4. The conserved motifs found in promoter of differentially expressed genes (DEGs).**

| Motif | E-value | Width | Best match in JASPAR | Significant GO term identified by GOMO |
|---|---|---|---|---|
| Motif 1 | 2.7e-346 | 29 | MA1281.1 | MF olfactory receptor activity |
| | | | | BP sensory perception of smell |
| | | | | BP DNA damage checkpoint |
| | | | | BP regulation of organ growth |
| | | | | MF proton-transporting ATPase activity, rotational mechanism |
| Motif 2 | 1.70E-295 | 31 | MA0631.1 | MF olfactory receptor activity |
| | | | | BP sensory perception of smell |
| Motif 3 | 8.00E-280 | 21 | MA0234.1 | CC mitochondrion |
| | | | | BP RNA metabolic process |
| | | | | MF structural constituent of ribosome |
| | | | | BP translational elongation |
| | | | | CC ribosome |
| Motif 4 | 4.2e-394 | 36 | MA0814.1 | MF histone binding |
| | | | | BP translational elongation |
| | | | | CC cytosolic ribosome |
| | | | | MF structural constituent of ribosome |
| | | | | BP sensory perception of smell |
| Motif 5 | 3.30E-220 | 33 | MA1242.1 | MF transcription factor activity |
| | | | | BP regulation of transcription from RNA |
| | | | | polymerase II promoter |
| | | | | BP RNA splicing |
| | | | | CC intracellular membrane-bounded organelle |
| Motif 6 | 6.40E-211 | 21 | MA1274.1 | MF olfactory receptor activity |
| | | | | BP sensory perception of smell |
| | | | | BP DNA damage checkpoint |
| | | | | BP regulation of organ growth |
| | | | | CC mitochondrion |
| Motif 7 | 5.70E-157 | 15 | MA1274.1 | MF olfactory receptor activity |
| | | | | BP sensory perception of smell |
| | | | | BP RNA processing |
| | | | | BP negative regulation of alpha-beta T cell differentiation |
| | | | | BP regulation of organ growth |
| Motif 8 | 6.00E-153 | 29 | MA0373.1 | MF olfactory receptor activity |
| | | | | BP sensory perception of smell |
| | | | | CC cytosolic ribosome |
| | | | | BP chemotaxis |
| | | | | BP translational elongation |
| Motif 9 | 1.90E-158 | 29 | MA0212.1 | MF structural constituent of ribosome |
| | | | | CC mitochondrial inner membrane |
| | | | | BP translational elongation |
| | | | | BP nuclear mRNA splicing, via spliceosome |
| | | | | CC intracellular organelle lumen |
| Motif 10 | 6.70E-184 | 27 | MA0299.1 | BP glucose catabolic process |
| | | | | MF olfactory receptor activity |
| | | | | BP sensory perception of smell |
| Motif 11 | 1.00E-177 | 40 | MA0505.1 | BP spliceosome assembly |
| | | | | BP DNA damage checkpoint |
| | | | | CC cytosolic ribosome |

**Gene dendrogram and module colors**

**Fig 3. Clustering dendrograms and modules identified by weighted gene co-expression network analysis (WGCNA).** The dendrogram indicates the gene clustering based on the TOM dissimilarity measure and each line indicated a gene. The colored column below the dendrogram indicates the modules conducted by the static tree cutting method at module size of 30 resulted in 8 color-coded modules.

TFs. The *EMP1*, *ELP5*, *ABCC3*, *PIGN*, *LTBP3*, *PANX1*, *DERL2* and *RPS5* genes in the modules indicated top-ranking in each module (Table 5).

## Survival analysis of hub genes

Survival analyses to analyze the correlation between hub gene expression and the prognosis of PDAC have been performed by using the GEPIA tool. *EMP1*(log-rank p = 0.008), *EVL* (log-rank p = 0.0076), *HSD17B14* (log-rank p = 0.00072), *MTERF4* (log-rank p = 0.0072), *RAB11-FIP3* (log-rank p = 0.002), and *TOM1L2* (log-rank p = 0.0016) were the only genes with significant overall survival (log-rank P < .05) (Fig 6).

## Discussion

Pancreatic ductal adenocarcinoma is a gastrointestinal malignant tumor that is diagnosed at an advanced stage due to a lack of effective screening tools and biomarkers. As a result, PDCA

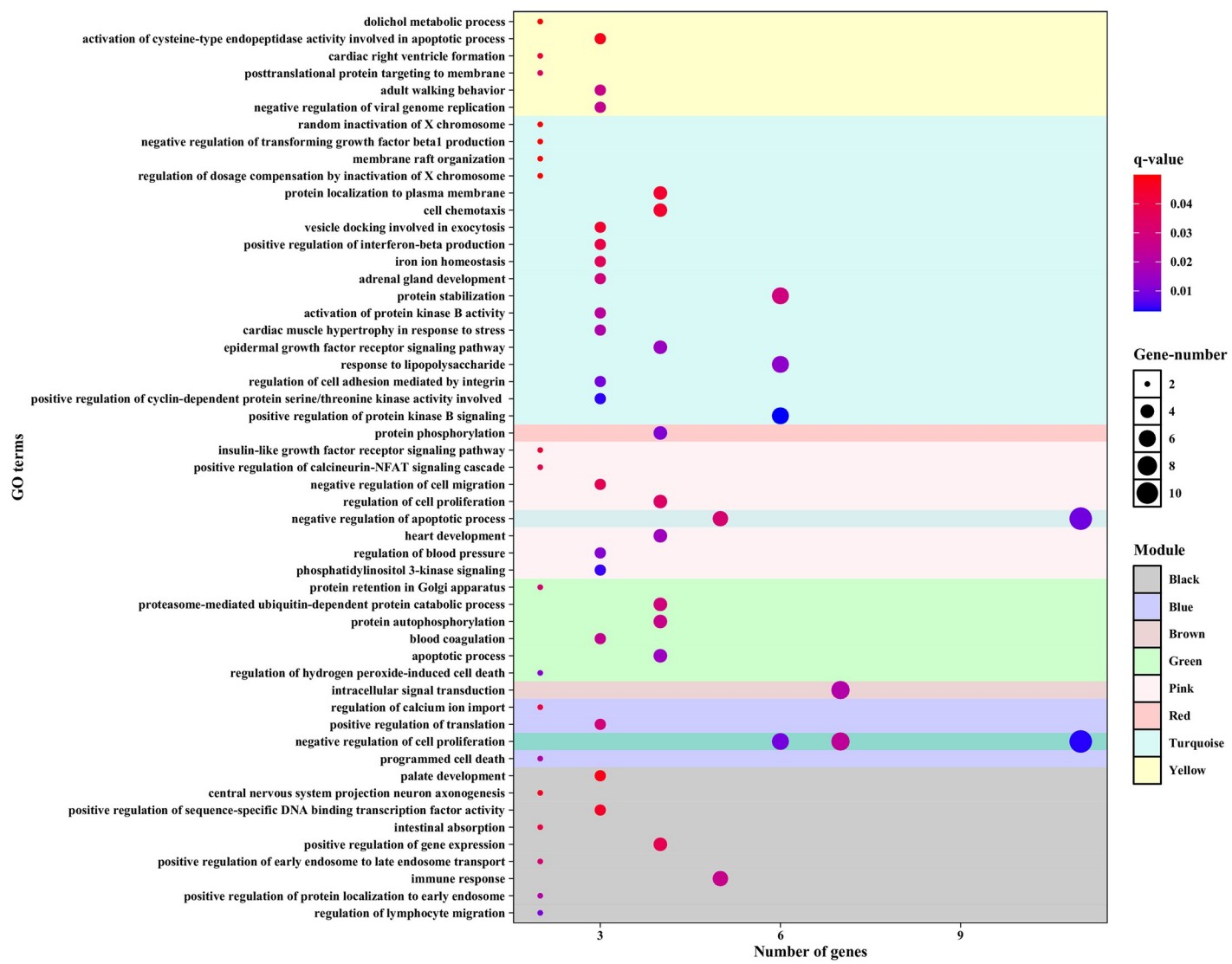

**Fig 4. Fifty-three significant biological processes were identified in eight modules with P-value<0.05.**

patients have a low survival rate. Therefore, the identification of reliable biomarkers associated with prognosis and treatment in PDCA is critical. There is a large amount of transcriptome data available, which allows researchers to identify biomarkers and metabolic pathways involved in various cancers [17]. In this regard, we collected 509 samples of microarray data from different datasets. Finally, 1074 DEGs were screened out by meta-analysis that had different expression levels in tumor tissues. The pathway enrichment analysis of DEGs showed that some enriched terms were related to the HIF-1 signaling pathway (KEGG:04066) and focal adhesion (KEGG:04510). HIF-1 has been reported to be involved in human cancers such as ovarian, prostate, and breast cancers. Deng et al., found that HIF-1 signaling increases in hepatocellular carcinoma compared to normal tissue and also plays a major role in cancer prognosis [18]. HIF-1α is a major factor involved in the regulation of cellular responses in prostate cancer; it is activated and decreased by hypoxia and targets the HIF pathway [19]. On the

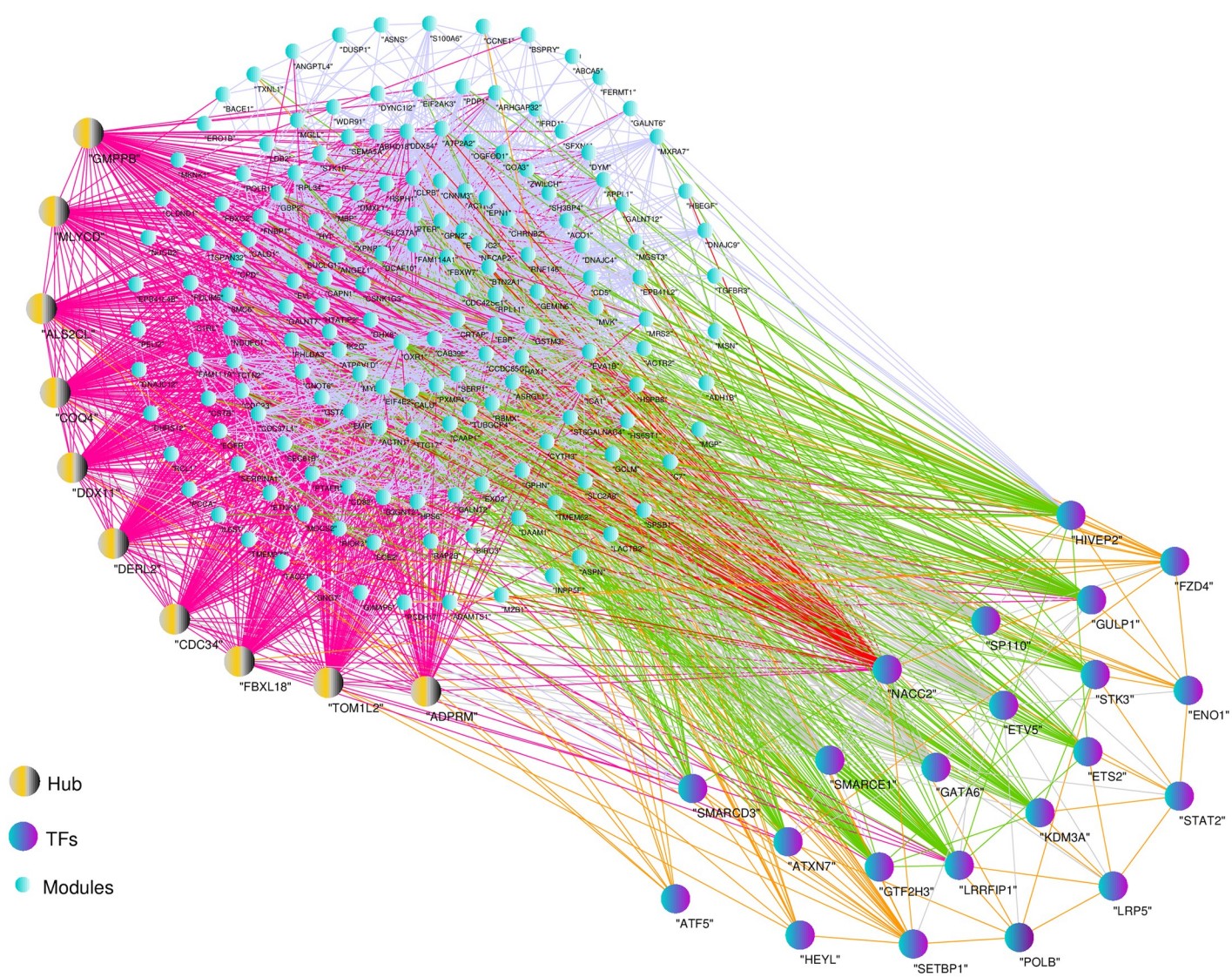

**Fig 5. Correlation of TFs with turquoise module.**

**Table 5. Top first hub for each module.**

| Hub gene | Module |
|----------|--------|
| *EMP1* | Black |
| *ELP5* | Blue |
| *ABCC3* | Brown |
| *PIGN* | Green |
| *LTBP3* | Pink |
| *PANX1* | Red |
| *DERL2* | Turquoise |
| *RPS5* | Yellow |

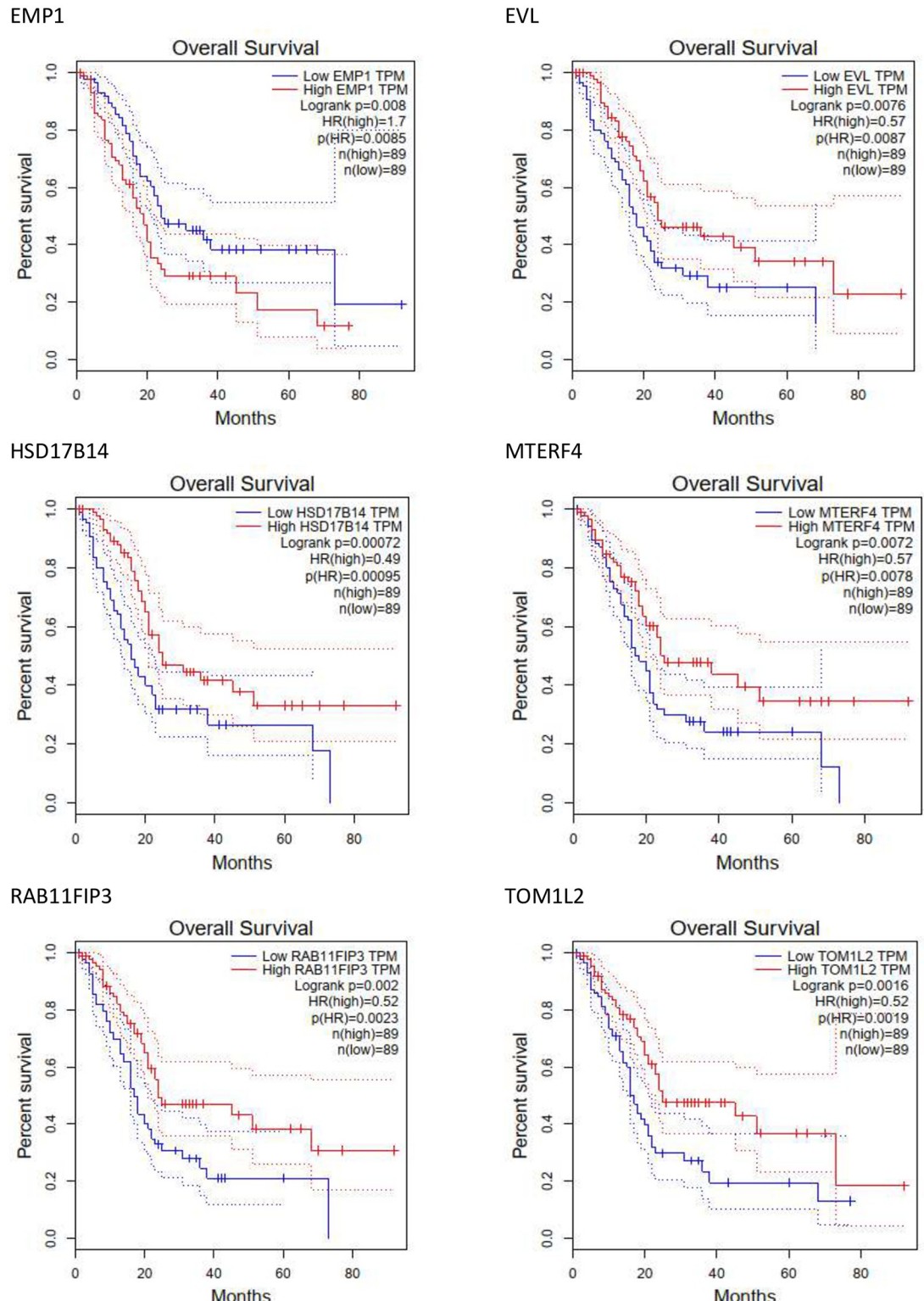

**Fig 6. Survival analysis of six signature genes based on GEPIA data.**

other hand, cancer cells consume oxygen, disrupt oxygen balance, and cause hypoxia, while cell growth and proliferation result in an increase in oxygen [20]. In fact, with increasing and decreasing oxygen levels, conditions are created for tumor growth and the survival of cancer cells increases [21]. However, oxygen in eukaryotic cells is essential for aerobic metabolism and ATP production. Therefore, it is important to maintain oxygen homeostasis. Focal adhesions play important roles in biological processes such as cell motility, cell proliferation, cell differentiation, regulation of gene expression, and cell survival [22]. They are communicators and adhesion between cells and the extracellular matrix (ECM). Focal adhesion kinase (FAK), a cytoplasmic non-receptor tyrosine kinase, is a key regulator in FAs, which leads to FA signals on cell adhesion to the ECM [23]. It has been reported that FAK is expressed in pancreatic cancer cell lines at the levels of mRNA, protein, and phosphorylated protein. It has previously been shown that FAK knockdown and FAK kinase inhibition have antitumor activity [24].

To study the regulatory mechanisms, we identified transcription factors among DEGs. The Cys2-His2 zinc finger family (C2H2-ZF) was the largest of the 152 TFs discovered. *Cys-His2* zinc finger (C2H2-ZF) proteins are the largest class of putative human transcription factors. Najafabadi et al. have indicated that the human genome contains an extensive and largely unexplored *C2H2-ZF* regulatory network that targets various genes and pathways [25]. Munro et al. have found that somatic mutations within *Cys2His2* zinc finger genes lead to widespread transcriptional dysregulation in cancer cells [26]. In the present study, *EGR1* was identified as one of the TFs that belong to the family of C2H2-ZF. *EGR1* is involved in tumor cell proliferation, invasion and metastasis, and tumor angiogenesis. It has also been reported that δ-tocotrinol in pancreatic cancer cells stimulates the expression of *EGR1*, which causes apoptosis of pancreatic cancer cells [27].

The results of meta-analysis were further investigated for protein kinases. Protein kinases are kinase enzymes that phosphorylate a target protein to change its function. Protein kinases have been shown in studies to be important cancer regulators [28]. In this study, we identified several protein kinase genes that associate with pancreatic cancer. Sixty-four protein kinases from 12 families were identified. The largest families included CAMK, TyR, STE and CMGC. STKs, which are members of the STE family, are enzymes that modulate protein activity by phosphorylating the serine and threonine amino acids [29]. This family is involved in signal transduction pathways, controlling metabolism, cell division, and angiogenesis [30]. They have also been shown to be involved in various types of cancer. For example, STK 4 is involved in pancreatic [31] and colorectal cancers [32, 33]. In addition, epidermal growth factor–containing fibulin-like extracellular matrix protein 1 (*EFEMP1*) as a member of the CMGCs family was detected.

*EFEMP1* is involved in anti-angiogenic via suppression of endothelial cell sprouting [34]. Changes in *EFEMP1* expression have previously been linked to cancers such as lung, liver, breast, prostate, nasopharyngeal, and pancreatic adenocarcinoma [35–41].

Considering the importance of miRNAs in cancer, we also identified miRNAs associated with DEGs. A total of 901 miRNA-related genes belong to 195 families, which may have important roles in pancreatic cancer. miRNAs can affect the expression profiles of genes such as oncogenes and tumor suppressor genes, as well as cause the formation and progression of cancer [42]. Hsa-miR-200 was the largest family, with 59 members. Peng et al. reported that the miR-200 family is involved in the onset and metastasis of cancer [43]. According to Barshack et al., the hsa-miR-200 family expression has been significantly increased in liver malignancies, which can be used in tumor diagnosis [44]. Moreover, Yu et al. suggested miR-200c as a new marker for the prognosis of pancreatic cancer [45].

We also performed a promoter analysis to identify regulatory elements upstream of the DEGs. Eleven motifs with significant scores were discovered. A large number of olfactory

receptor activity–associated motifs were detected in the DEG upstream promoter sequences. Olfactory receptors (ORs) are a large group of G protein-coupled receptors in the olfactory epithelium [46]. ORs are expressed ectopically in many tissues, and some evidence points to the role of ORs in several diseases, including cancer [47]. They are also involved in various physiological processes such as cell migration, proliferation, and secretion [48]. They have also been mentioned in several studies as biomarkers for various cancer tissues such as breast cancer [47, 48], bladder cancer [46], and small intestine neuroendocrine carcinomas [49].

WGCNA was performed using the DEGs obtained from meta-analysis to identify the co-expressed and hub genes, and a total of 8 modules were discovered. From each module, ten hub genes were extracted. Based on GO enrichment analysis, the hub genes act as regulators of cell proliferation as well as regulators of apoptosis and cell death. Eleven of the hub genes (*EMP1*, *EVL*, *ELP5*, *DEF8*, *MTERF4*, *GUlP1*, *CAPN1*, *IGF1R*, *HSD17B14*, *TOM1L2* and *RAB11FIP3*) were related to apoptosis, suggesting that the genes may be a potential target in PDCA. The survival analysis of these 11 genes revealed that *EMP1*, *EVL*, *HSD17B14*, *MTERF4*, *RAB11FIP3*, and *TOM1L2* expression are closely related to the prognosis of patients with PDAC. It was found that *EMP1* (epithelial membrane protein 1) encodes a protein located in the cell membrane. This gene is involved in biological processes such as cell death, epidermal development, and bubble assembly [50]. In a study by Liu et al., *EMP1* was reported to be expressed in a large number of tumors and was shown to be a cellular linkage on cell membranes and to play an important role in proliferation, invasion, metastasis of tumor cells, and mesenchymal epithelial transmission. Furthermore, *EMP1* has been shown in several studies to be a reliable biomarker in cancers such as gastric [51], colorectal [52] ovarian [53], bladder urothelial carcinoma [54] and non-small lung carcinoma [55]. *EVL* is a member of the Ena / VASP family of proteins involved in the regulation of the actin cytoskeleton. Changes in cytoskeletal composition either stimulate or suppress tumor cell invasion and migration. Mouneimne et al. showed that *EVL* decreased the migration and invasion of tumor cells. Decreased *EVL* expression in human tumor cells is also associated with high invasive activity, increased protrusion, decreased contraction and adhesion [56]. This gene is also involved in cervical cancer [57]. The *HSD17B14* gene encodes 17β- Hydroxysteroid dehydrogenase. Sivik et al. discovered that *HSD17B14* is a predictor marker for the tamoxifen response in breast cancer [58]. The *MTERF* family includes *MTERF1*, *MTERF2*, *MTERF3*, and *MTERF4* that have roles in the pathogenesis of various cancer types. In a study by Sun et al., it was indicated that high mRNA expression levels of the *MTERF* family lead to an improved overall survival (OS) rate in patients with lung adenocarcinoma. Furthermore, in their study, they identified *MTERFs* as primary biomarkers for predicting non-small cell lung cancer [59]. *RAB11FIP3* is an interaction of RAB11GTPase with the FIP3 protein. RAB11 GTPase is a major regulator of vesicle trafficking and belongs to a family of proteins that are susceptible to changes in human cancers [60]. Tong et al. showed that *RAB11FIP3* is involved in the endocytosis recycling in breast cancer and promotes EGFR transmission [61]. The next gene in the list, Tom1l2, belongs to the Tom1 family that may be involved in the immune response and suppression of tumors [62].

## Conclusions

In conclusion, in this study, several bioinformatics methods were used to identify novel biomarkers for pancreatic cancer. 1074 DEGs were screened and TFs, PKs, miRNAs, and regulatory elements were identified by analysis. Following that, among the DEGs, 11 important hub genes were found that were associated with many pathways of tumor progression. Among them, *EMP1* and *RAB11FIP3* were identified as new biomarkers for the treatment and

prognosis of patients with PDAC. A comprehensive study is required in future research to confirm the prognostic and diagnostic value of the identified biomarkers. Therefore, they may be promising prognostic indicators for patients with PDAC.

## Supporting information

**S1 Fig.**
(PDF)

**S1 Table.**
(XLSX)

**S2 Table.**
(XLSX)

**S3 Table.**
(XLSX)

**S4 Table.**
(XLSX)

**S5 Table.**
(XLSX)

**S6 Table.**
(XLSX)

**S7 Table.**
(XLSX)

**S8 Table.**
(XLSX)

**S9 Table.**
(XLSX)

**S10 Table.**
(XLSX)

**S11 Table.**
(XLSX)

**S12 Table.**
(XLSX)

## Author Contributions

**Conceptualization:** Shirin Omidvar Kordshouli, Ahmad Tahmasebi, Amin Ramezani, Ali Niazi.

**Data curation:** Shirin Omidvar Kordshouli.

**Formal analysis:** Shirin Omidvar Kordshouli.

**Investigation:** Ahmad Tahmasebi.

**Methodology:** Shirin Omidvar Kordshouli, Ali Moghadam.

**Supervision:** Ali Niazi.

**Visualization:** Shirin Omidvar Kordshouli.

**Writing – original draft:** Shirin Omidvar Kordshouli, Ahmad Tahmasebi, Amin Ramezani.

**Writing – review & editing:** Ahmad Tahmasebi, Ali Moghadam, Amin Ramezani, Ali Niazi.

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
