## [Decision Letter · Decision Letter 0]

2 May 2023

PONE-D-23-02176A comprehensive meta-analysis of transcriptome data to identify signature genes associated with pancreatic ductal adenocarcinomaPLOS ONE

Dear Dr. Niazi,

Thank you for submitting your manuscript to PLOS ONE. After careful consideration, we feel that it has merit but does not fully meet PLOS ONE’s publication criteria as it currently stands. Therefore, we invite you to submit a revised version of the manuscript that addresses the points raised during the review process.

We look forward to receiving your revised manuscript.

Kind regards,

Surinder K. Batra

Academic Editor

PLOS ONE

Reviewers' comments:

Reviewer's Responses to Questions

**Comments to the Author**

1. Is the manuscript technically sound, and do the data support the conclusions?

Reviewer #1: No

Reviewer #2: Yes

2. Has the statistical analysis been performed appropriately and rigorously? 

Reviewer #1: I Don't Know

Reviewer #2: Yes

3. Have the authors made all data underlying the findings in their manuscript fully available?

Reviewer #1: Yes

Reviewer #2: No

4. Is the manuscript presented in an intelligible fashion and written in standard English?

Reviewer #1: Yes

Reviewer #2: Yes

5. Review Comments to the Author

Reviewer #1: The authors did not present clinical characteristics of samples used for analysis. The analysis has to be repeated with consideration to stages of PDAC, preferably contrasting early-stage vs. late-stage cases. If only late-stage samples were used - the results have little translational relevance since the identified DEGs could unlikely serve as therapy targets.

Reviewer #2: The study identified several novel biomarkers for pancreatic ductal adenocarcinoma using a meta-analysis. The authors did a great work preforming several bioinformatics analyses to identify novel biomarkers.

Minor comments/edits and questions:

- The authors mentioned several abbreviations (TFs, PKs, CREs, some others) the first time in the main text, I would advise the authors to include the full name.

- The authors included the P-value for the DEGs in the supplementary but the txt mention that the filtering criteria was the FDR value. Does that mean all the 1074 genes are statistically significant using both pvalue and FDR? What are the log2FoldChange and padj values for those DEGs?

- A total of 39 up and 25 down PKs were regulated, do you mind including the list in the supplementary?

- There are a total of 901 miRNAs, and they belonged to 195 families, do you mind including the list in the supplementary?

- For the WGCNA, the different colors have different number of genes, what are the number of genes for all the different groups? Do you mind providing the full list with their genes?

- For enrichment analysis (DAVID), a p-value was used to find the significant. Why the padj value wasn’t used same as the GO analysis? Also, Fig #4, says q-value but the text says the p-value was used. Which one was used in this case?

- The authors mentioned “the highest number of TFs belonged to the turquoise module”, wouldn’t that be always true because it has the highest number of genes among the other modules?

- End page #7, There are 11 PKs genes in the turquoise module, do you mind including the list?

- What are those enriched 53 GO and 25 pathways for the hub genes?

- Page #8, one of the in-text citation is not consistent with the others.

- Page #11, the author mention “these genes” in the first line, do you mean the hub genes?

6. PLOS authors have the option to publish the peer review history of their article (what does this mean?). If published, this will include your full peer review and any attached files.

Reviewer #1: No

Reviewer #2: No

---

## [Author Response · Author response to Decision Letter 0]

12 Jun 2023

Response to Reviewer 1

Reviewer #1: The authors did not present clinical characteristics of samples used for analysis. The analysis has to be repeated with consideration to stages of PDAC, preferably contrasting early-stage vs. late-stage cases. If only late-stage samples were used - the results have little translational relevance since the identified DEGs could unlikely serve as therapy targets.

Comment: Many thanks to the reviewer for taking the time to read and comment on our manuscript.

We appreciate the insightful comment provided by the reviewer and would like to address their concerns regarding the clinical characteristics of the samples used in our meta-analysis. Our primary objective in this study was not solely focused on the identification of genes with therapeutic potential. However, unfortunately complete clinical characteristics were not available for all the samples included in the meta-analysis. While we acknowledge the significance of considering clinical characteristics in the study of PDAC, our meta-analysis primarily aimed to identify signature genes associated with the disease. This was achieved by increasing the effective sample size and obtaining a more accurate estimate of the effect size, contributing to a better understanding of the molecular mechanisms underlying PDAC. To achieve this objective, we employed an approach that involved aggregating data from multiple studies. Meta-analysis, a widely accepted method, combines data from various sources to enhance statistical power and improve the generalizability of findings. It is worth noting that the use of meta-analysis to identify biomarkers and molecular signatures has been applied in numerous studies across different cancer types. For instance, studies such as "A meta-analysis of gene expression-based biomarkers predicting outcome after tamoxifen treatment in breast cancer," "Meta-analysis-based gene expression profiling reveals functional genes in ovarian cancer," and "Meta-Analysis of Microarray Expression Studies on Metformin in Cancer Cell Lines" aimed to identify significant molecular features without specifically considering clinical characteristics. However, to address concerns about non-biological variation and data heterogeneity, we performed batch effects correction among the different datasets included in our meta-analysis. By employing a Bayesian experimental method, we successfully removed batch effects and improved data homogeneity, thereby ensuring the robustness and reliability of our findings. Overall, we believe that our study contributes to a broader understanding of the molecular landscape of PDAC and provides valuable insights for further research in this field.

 

Response to Reviewer 2

Reviewer #2: The study identified several novel biomarkers for pancreatic ductal adenocarcinoma using a meta-analysis. The authors did a great work preforming several bioinformatics analyses to identify novel biomarkers.

Minor comments/edits and questions:

Comment: We appreciate the reviewer’s positive and constructive comments. The followings are our point-by-point responses:

The authors mentioned several abbreviations (TFs, PKs, CREs, some others) the first time in the main text, I would advise the authors to include the full name. 

The correction is made according to the reviewer's comment.

The authors included the P-value for the DEGs in the supplementary but the txt mention that the filtering criteria was the FDR value. Does that mean all the 1074 genes are statistically significant using both pvalue and FDR? What are the log2FoldChange and padj values for those DEGs?

Thank you for letting us know. All 1074 genes are statistically significant using FDR. This has also been corrected in the supplementary file.

A total of 39 up and 25 down PKs were regulated, do you mind including the list in the supplementary? 

Following the reviewer's comment, the list of up/down regulated PKs is added to the revised manuscript and Supplementary file.

There are a total of 901 miRNAs, and they belonged to 195 families, do you mind including the list in the supplementary?

In accordance with the reviewer's comment, the list of miRNAs is added to the revised manuscript and the supplementary file.

For the WGCNA, the different colors have different number of genes, what are the number of genes for all the different groups? Do you mind providing the full list with their genes?

A complete list of genes for each module has been generated and included in the supplementary file provided with the revised manuscript.

For enrichment analysis (DAVID), a p-value was used to find the significant. Why the padj value wasn’t used same as the GO analysis? Also, Fig #4, says q-value but the text says the p-value was used. Which one was used in this case?

Thank you for letting us know. We used q-value to identify significant GO terms. This will also be corrected in the revised manuscript.

The authors mentioned “the highest number of TFs belonged to the turquoise module”, wouldn’t that be always true because it has the highest number of genes among the other modules?

Yes, this is correct. Considering that this module has the largest number of genes among other modules, it is more likely to contain more TFs. However, the purpose of naming this set is to emphasize the regulatory role of this module.

End page #7, There are 11 PKs genes in the turquoise module, do you mind including the list?

According to the reviewer's comment, the list of PKs of the turquoise module is added to the revised manuscript and the supplementary file.

What are those enriched 53 GO and 25 pathways for the hub genes?

The list of Go terms and pathways is added to the revised manuscript and the supplementary file.

Page #8, one of the in-text citations is not consistent with the others. 

Thanks for your comment. The article is added to the references section.

Page #11, the author mention “these genes” in the first line, do you mean the hub genes?

In the revised manuscript, "these genes" has been changed to "the hub genes".

---

## [Decision Letter · Decision Letter 1]

21 Jul 2023

A comprehensive meta-analysis of transcriptome data to identify signature genes associated with pancreatic ductal adenocarcinoma

PONE-D-23-02176R1

Dear Dr. Niazi,

We’re pleased to inform you that your manuscript has been judged scientifically suitable for publication and will be formally accepted for publication once it meets all outstanding technical requirements.

Kind regards,

Surinder K. Batra

Academic Editor

PLOS ONE

Additional Editor Comments (optional):

Reviewers' comments:

Reviewer's Responses to Questions

**Comments to the Author**

1. If the authors have adequately addressed your comments raised in a previous round of review and you feel that this manuscript is now acceptable for publication, you may indicate that here to bypass the “Comments to the Author” section, enter your conflict of interest statement in the “Confidential to Editor” section, and submit your "Accept" recommendation.

Reviewer #1: All comments have been addressed

Reviewer #2: All comments have been addressed

2. Is the manuscript technically sound, and do the data support the conclusions?

Reviewer #1: Yes

Reviewer #2: Yes

3. Has the statistical analysis been performed appropriately and rigorously? 

Reviewer #1: Yes

Reviewer #2: Yes

4. Have the authors made all data underlying the findings in their manuscript fully available?

Reviewer #1: Yes

Reviewer #2: Yes

5. Is the manuscript presented in an intelligible fashion and written in standard English?

Reviewer #1: Yes

Reviewer #2: Yes

6. Review Comments to the Author

Reviewer #1: The authors have performed a comprehensive meta-analysis of transcriptome data to identify biomarkers associated with pancreatic cancer. Eleven novel hub genes associated with tumor progression have been discovered including EMP1 and RAB11FIP3. The study is solid and offers initial discovery for the follow-up clinical validation. The authors have addressed critiques from the previous reviewers.

Reviewer #2: The authors did a great work addressing the comments.

Only two minor edits are needed:

- The authors mentioned CREs for the first time in the main text, I would advise the authors to include the full name.

- Table S4 should reference the following sentence ("A total of 39 and 25 PKs were up- and down-regulated, respectively.")

7. PLOS authors have the option to publish the peer review history of their article (what does this mean?). If published, this will include your full peer review and any attached files.

Reviewer #1: No

Reviewer #2: No

---

## [Editor Report · Acceptance letter]

27 Jul 2023

PONE-D-23-02176R1 

A comprehensive meta-analysis of transcriptome data to identify signature genes associated with pancreatic ductal adenocarcinoma 

Dear Dr. Niazi:

I'm pleased to inform you that your manuscript has been deemed suitable for publication in PLOS ONE. Congratulations! Your manuscript is now with our production department. 

Kind regards, 

on behalf of

Prof. Surinder K. Batra 

Academic Editor

PLOS ONE